# Collagen Type-I Agent Reduced Postoperative Bowel Adhesions Following Laparoscopic and Robot-Assisted Radical Prostatectomy: A Prospective, Single-Blind Randomized Clinical Trial

**DOI:** 10.3390/jcm11175058

**Published:** 2022-08-28

**Authors:** Seokhwan Bang, Young Hyo Choi, Seung-Ju Lee, Sung-Hoo Hong

**Affiliations:** 1Department of Urology, Seoul St. Mary’s Hospital, Collage of Medicine, The Catholic University of Korea, Seoul 06591, Korea; 2Department of Urology, St. Vincent’s Hospital, Collage of Medicine, The Catholic University of Korea, Seoul 16247, Korea

**Keywords:** postoperative adhesion, laparoscopy, robotic surgery, adhesion barrier

## Abstract

This study aimed to compare the anti-adhesive effect of collagen type-I (COL) agent and hyaluronic acid-carboxymethylcellulose (HA/CMC) following laparoscopic and robotic radical prostatectomies. This study was a randomized, controlled, single-blind, multicenter study using COL and HA/CMC in patients who underwent laparoscopic and robotic radical prostatectomies. All patients were randomly assigned to either the COL (n = 66) or HA/CMC (n = 65) group. Viscera slide ultrasound sonography was recorded on the day of surgery (V2) and 12 weeks later (V4). The primary end point was the difference in the excursion distance in the viscera slide ultrasonography between V2 and V4. A total of 131 patients participated in this study. The viscera slide distance in the test and control groups was 1.89 ± 0.49 cm and 1.80 ± 0.45 cm, respectively, at V2 (*p* = 0.275). The average distance of the viscera slide in the test and control group was 1.59 ± 0.49 cm and 1.64 ± 0.45 cm, respectively at V4 (*p* = 0.614). None of the patients showed significant adverse effects. This randomized study showed that the efficacy and stability of the gel-type COL anti-adhesion agent are not inferior to those of HA/CMC, of which the properties are established.

## 1. Introduction

Adhesion of organs and tissues that occurs after surgery is a natural process in which damaged tissue cells proliferate; however, excessive adhesions or unintentional adhesions with other organs can lead to organ dysfunction and can be life-threatening. Peritoneal adhesions most frequently occur from peritoneal damage caused by abdominal and pelvic surgeries [1,2]. It arises from an intra-abdominal infection or abdominal trauma. Few studies report that patients develop adhesions in approximately 93–100% of abdominal open surgeries due to ischemia, thermal injury, infections, and residual blood clot during the surgery [3].

Adhesions occurring in the abdominal cavity are the most common cause of small bowel obstruction (SBO) and are reported as a cause in approximately 65–75% of cases of small intestine obstruction [3]. Small intestine obstruction caused by intra-abdominal adhesions is likely to cause infertility, chronic abdominal pain, prolonged duration of surgery, and increased costs. It usually occurs after several months to a few years after surgery, due to which it may not be detected. Prevention and reduction of adhesion is the best strategy to handle it [4].

Laparoscopic surgery is a surgical method that minimizes wounds during surgery. Although laparoscopic surgery is a common procedure, there is still a risk that intra-abdominal structures or organs can be damaged during trocar insertion, thereby limiting the prevention of adhesion. Therefore, adhesion prevention using various drugs or agents have been attempted, and recently anti-adhesion agents using physical barriers have been developed. In case of anti-adhesion using drugs, while the effect of interfering with the fibrin deposition has the disadvantage of delaying healing, a physical barrier creates a potential for adhesion between the two adjacent tissues. It is widely used in clinical practice because it is located between the surfaces and can separate tissues. As a material for physical barriers made from naturally occurring polysaccharides, oxidized regenerated cellulose, sodium carboxylmethyl cellulose (CMC), dextran, sodium hyaluronate (HA), polyethylene glycol (PEG), poloxamer, and gore-tex are known as synthetic polymers. During the healing period of the damaged tissue, it acts to prevent adhesion, after which it decomposes or gets absorbed by surrounding tissues. Temporarily creating a physical barrier in the abdominal cavity after surgery is known as an effective way to prevent adhesions, and HA/CMC is a well-known material [5]. Collagen is a major component of the extracellular matrix, and it is known that esterified collagen has a net positive charge which increases cell proliferation and enhances cell attachment [6]. 

Anti-adhesion agents that are currently used are classified as solutions, gels, films, or membranes. Solution or gel-type anti-adhesion agents are easy to apply and have high viscosity. However, they are slurred from the wound due to gravity, and it is difficult to control the duration of the physical barrier maintenance to prevent adhesions because the process of absorption and excretion in the body occurs rapidly. On the other hand, in the case of a film or membrane-type material, it can be used in a relatively wide surgical area, but depending on the product characteristics, it is difficult to repair the wound by suturing and difficult for the surgeon to attach it perfectly to the desired position.

In this study, we tried to prove the role of the gel-type collagen-I agent (Collabarrier^®^) as an anti-adhesive agent by performing a non-inferiority test by comparing hyaluronic acid-carboxymethylcellulose (HA/CMC) agent (Guardix-sol^®^) which have already proven their effectiveness. 

## 2. Materials and Methods

This was a prospective, randomized, multicentered, single-blind controlled study using Collabarrier^®^ (Dalimtissen, Seoul, Korea) and Guardix-sol^®^ (Genewell, Seoul, Korea) in patients who underwent robot-assisted radical prostatectomy or laparoscopic radical prostatectomy between July 2019 and May 2021 in St. Mary’s hospital (Seoul, Korea) and St. Vincent’s hospital (Suwon, Korea). All patients were informed in detail about the aims and procedures of the study, and they signed a written informed consent prior to inclusion in the study. The protocol and the written informed consent were approved by the local ethical committee (the Catholic University of Korea Seoul St Mary’s Hospital IRB [KC19DDSE0230] and the Catholic University of Korea, St. Vincent’s Hospital IRB [No VC20DDDT0078]).

### 2.1. Participants

Patients of >20 years of age who were diagnosed with prostate cancer and scheduled to undergo robot-assisted radical prostatectomy or laparoscopic radical prostatectomy were eligible. Exclusion criteria included those who have hypersensitivity or allergic reactions to clinical investigational devices, infection or contamination at the surgical site, previous abdominal surgery, uncontrolled diabetes, and a history of severe drug allergy, including those with a history of medical conditions causing intestinal adhesions.

### 2.2. Study Design and Protocol

The laparoscopic radical prostatectomy and robot-assisted radical prostatectomy were performed by two expert urologists (SHH and SJL) in two medical centers (Seoul St. Mary’s medical center, Seoul, Korea and St. Vincent’s Hospital, Suwon, Korea). Both surgeons have experience in over 150 cases of robotic and laparoscopic radical prostatectomies. They followed and performed the same surgical procedure during the clinical trial. The robotic and laparoscopic radical prostatectomies were performed by a fan-shaped transperitoneal approach with five trocars. All robotic surgeries were performed with Da Vinci Xi model (Instiutive, Sunnyvale, CA, USA). All robotic surgery was used for four 8 mm trocar, and one 11 mm trocar was used for assist. Laparoscopic surgery used five 11 mm trocar. After the docking of robots, we approached the peritoneal cavity, incising the parietal peritoneum between the medial umbilical ligaments, which were dissected through the fatty alveolar tissue to develop the space of Retzius. The surgical steps were as follows: (1) incision of the endopelvic fascia; (2) suturing of the dorsal vein complex; (3) division of the bladder neck; (4) dissection of the seminal vesicles; (5) dissection of the Denonvillier fascia and division of the lateral pedicles with antegrade neurovascular bundle dissection; (8) apical dissection and division of the dorsal vein and urethra; and (9) anastomosis of urethra and bladder neck [7].

This study included 120 participants (60 per group) considering a 10% dropout rate. The sample size was determined with a level of significance α = 0.05 (two-sided) and 80% statistical power of the test. All patients were randomly assigned to either the collagen type-I treatment group or the HA/CMC treatment group using a computer-generated randomization table. The patients were blinded to the randomized group. Based on the group, fibrin or HA/CMC was applied in all port sites and the peritoneal incision line of the medial umbilical ligament with a single-use applicator. The amount of each agent was 5 mL. All medical information was collected during enrollment (V1). Viscera slide ultrasound was recorded during (V2) and 12 weeks (V4) after the surgery. Adverse effects were recorded during (V2), and one (V3) and 12 weeks (V4) after the surgery. The primary end point was the difference in excursion distance on viscera slide ultrasound between V2 and V4. 

### 2.3. Evaluation of Safety and Efficacy

Bowel adhesion was evaluated on the day of the surgery and 12 weeks postoperatively. We performed ultrasound sonography according to a previously described technique [8]. Independent evaluators observed the viscera slide through ultrasound. The patient was instructed to breathe normally; the probe was placed around the umbilicus in the longitudinal direction, and the examination was started. Subsequently, the patient was instructed to exaggerate breathing, and the viscera slide was measured. The movement distance in the longitudinal direction was measured by observing a clear high echoic area in the visceral image observed during the patient’s continuous breathing process. The observation area of the viscera slide was divided according to the insertion site of the trocar; observation was performed for each site, and the average value of the distance of the viscera slide measured for each site was used for the evaluation of effectiveness. Normal viscera sliding movement was defined as ≥1 cm of longitudinal movement. Restricted viscera slide was defined as <1 cm of longitudinal movement during both normal and exaggerated respiration. While two or more independent evaluators at one institution interpreted the results, the independent central evaluator divided the clinical trial participants and evaluated them separately. Ultrasound examinations of V2 and V4 were performed by the same independent central evaluator per clinical trial participant, and the reliability was secured through a cross-examination with the findings of the independent central evaluator.

### 2.4. Statistical Analysis

For the analysis of demographic and other underlying characteristics in this clinical trial, independent two-sample t-test or Wilcoxon rank sum test was used according to the normality for continuous data, and chi-squared test was used for categorical data. If there were cells with an expected frequency of <5, Fisher’s exact test was used. *p* < 0.05 was considered statistically significant. Statistical calculations were performed with IBM SPSS statistics, version 21 (IBM Corp, Armonk, NY, USA) software.

## 3. Results

Among the patients diagnosed with prostate cancer who needed radical prostatectomy using laparoscopic or robotic surgery, 150 patients who voluntarily agreed with written consent to participate in the clinical trial were screened. A total of 134 patients enrolled in this clinical trial after screening, of which 66 were randomized to the test group and 68 to the control group. A total of 131 subjects were selected, excluding 3 patients of the control group who dropped out, and the study was conducted with 66 patients in the test group and 65 in the control group. 

The characteristics of the patients who participated in the study are summarized in Table 1.

Table 2 shows the results of the adhesion characteristics.

During V2, the viscera slide distance of the test and control groups was 1.89 ± 0.49 cm and 1.80 ± 0.45 cm, respectively, and there was no statistically significant difference (*p* = 0.275). Further, in V4, the average excursion distance of the viscera slide of the test and control groups was 1.59 ± 0.49 cm and 1.64 ± 0.45 cm, respectively, showing no statistically significant difference (*p* = 0.614). 

In the safety evaluation of this clinical trial, serious adverse events occurred in two patients in the test group and none in the control group. The two serious adverse events that occurred in the test group were reported as serious adverse events because they were ‘moderate’ and required hospitalization or extension of the hospitalization period but were evaluated as not related to test materials (Table 3).

## 4. Discussion

Through this clinical trial, we confirmed that there was no difference in the postoperative visceral excursion distance between the previously used HA/CMC anti-adhesion agent and the newly developed gel-type collagen type-I anti-adhesion agent. These results suggest that the anti-adhesion ability of the collagen type-I agent is not different from that of HA/CMC. Further, there was no difference between the two groups in terms of stability.

In previous studies, it was reported that all three types of ionized collagen (membrane, film, and gel) prevented adhesion based on animal experiments [9]. However, this study is the first to confirm the anti-adhesion ability of gel-type collagen after surgery through clinical trials. Collagen is one of the main components of the extracellular matrix and has been evaluated as the most suitable candidate as implant material, [10] of which the efficacy has been proven in this study.

The type of anti-adhesion agent is also an important factor. Anti-adhesion agents can be classified into solutions, gels, films, and membranes. Anti-adhesion agents in the form of films or membranes can be used in relatively wide surgical sites, but depending on the characteristics of the product, it may pose difficulty in suturing and fixing the wound area or handling it when undertaking surgeries in internal organs, due to which it may be difficult to accurately repair the wound area. In the case of an anti-adhesion agent in the form of a solution or gel, it is easy to apply and has a high viscosity, but there are disadvantages such as the difficulty in controlling the duration of the physical barrier maintenance to prevent adhesion because it flows down from the wound by gravity or the absorption and discharge process in the body occurs rapidly. The adhesive strength against gravity is one of the important factors for the anti-adhesion agent because the main preventive space for the anti-adhesion agent is the abdominal wall. Park et al. reported that liquid formulations such as gels have more advantages than membrane-type HA/CMC [11]. Additionally, in the case of laparoscopic or robotic surgery, which is widely used these days, considering that the wound window is not as wide compared to that in open surgery and the anti-adhesion agent can be sprayed in a relatively narrow space, the gel-type collagen anti-adhesion agent is considered excellent.

In this study, we evaluated the adhesion of the abdominal wall using ultrasound. This is the method introduced by Sigel B et al. [8]; ultrasound examination is a specific and reliable method to identify and detect adhesion-free areas [12,13]. In urological surgery such as prostatectomy, where the pelvic cavity is the main space and the adhesion target is the bowel, ultrasound can be used as the evaluation device as that in our study.

This study has several limitations. First, the number of participants in this study is not sufficient. Second, this study did not prove the absolute stability of the collagen agent, but rather demonstrated its effectiveness through comparison with the existing HA/CMC agent. Third, since this study was conducted for prostatectomy, it is not applicable to women. It seems necessary to prove the same efficacy in the abdominal cavity of women in future. Finally, in this study, we did not distinguish between robotic and laparoscopic surgeries. Since there is a clear difference between robotic and laparoscopic surgeries, it will be necessary to analyze this in future studies. Through a future study, we suggested a method for reducing adhesions in laparoscopic and robotic surgery, which is being gradually expanded. It is expected that these various attempts will contribute to creating a safer surgical environment and patient recovery.

## 5. Conclusions

This randomized study showed that the efficacy and stability of the gel-type collagen type-I anti-adhesion agent are not inferior to those of HA/CMC with proven efficacy and stability.

## Figures and Tables

**Table 1 jcm-11-05058-t001:** Demographic data of collagen type-I agent group (Test group) and hyaluronic acid carboxymethylcellulose (HA/CMC) group (Control group).

Variables	Test Group	Control Group	Total	*p*-Value
	(n = 66)	(n = 65)	(n = 131)	
Age (years), mean ± SD	67.24 ± 7.60	67.45 ± 6.07	67.34 ± 6.86	0.865
Height (cm), mean ± SD	166.65 ± 600	167.90 ± 5.65	167.27 ± 5.84	0.222
Weight (kg). mean ± SD	67.47 ± 9.74	67.96 ± 9.38	67.71 ± 9.53	0.768
Combined disease, n (%)				0.615
DM	8	14	22	
Cardiovascular disease	5	5	10	
Gastrointestinal disease, hepatitis	2	0	6	
Smoking history				0.313
Current smoker	8	12	20	
Non-smoker	58	53	111	
*p*-value was calculated by independent *t* test				
SD: standard deviation, DM: diabetes mellitus				

There was no statistically significant difference between each patient group.

**Table 2 jcm-11-05058-t002:** Adhesion characteristics in the test and control groups.

	Test Group (n = 66)	Control Group (n = 65)	*p*-Value
Ultrasound findings			
Average excursion distance of the viscera slide (cm) at V2	1.89 ± 0.49	1.80 ± 0.45	0.275
Average excursion distance of the viscera slide (cm) at V4	1.59 ± 0.49	1.64 ± 0.45	0.614
Difference between V2 and V4 (cm)	0.29 ± 0.29	0.16 ± 0.27	0.996
Number of restricted viscera slide sites (%)	5 (7.58)	5 (7.69)	1.000
*p*-value was calculated by independent *t* test			
V2: visit 2 (day of surgery), V4: visit 4 (12 weeks postoperatively)			

**Table 3 jcm-11-05058-t003:** Significant adverse effects in the two groups.

	Test Group (n = 66)	Control Group (n = 65)	Total (n = 131)
Total (n,%)	2 (3.03%)	0 (0.00%)	2 (1.53%)
Hepatobiliary disorders			
Acute cholecystitis	1 (1.52%)	0 (0.00%)	1 (0.76%)
Infections and infestations			
Pelvic abscess	1 (1.52%)	0 (0.00%)	1 (0.76%)

## Data Availability

Not applicable.

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
