# Peer review of "Collagen Type-I Agent Reduced Postoperative Bowel Adhesions Following Laparoscopic and Robot-Assisted Radical Prostatectomy: A Prospective, Single-Blind Randomized Clinical Trial"

_jcm, 2022, doi:10.3390/jcm11175058_

Round 1
Reviewer 1 Report
It Is a prospective, randomized, multicentered study comparing the Collagen type-I and the Hyaluronic acid-carboxymethylcellulose as bowel anti-adhesion agent. Based on the results, there is no significant difference as for the efficacy and the stability of the two agents.
Comment 1: I believe that in the introduction section the anti-adhesive mechanism of the two compared agents has to be included.
Comment 2: In my opinion there is a lack of demographic characteristics. It is essential for the presence/absence of lymph node dissection and post-operative complication to be mentioned as they may increase the risk of post-operative bowel adhesions.
Comment 3: As the authors mention in the limitations, there are no subgroups of the patients who underwent laparoscopic radical prostatectomy and robotic-assisted radical prostatectomy. Even if the two subgroups are not well-balanced, some preliminary results about the efficacy of the two anti-adhesive agents in every subgroup could be very interesting.
Comment 4: In table I the two groups are not mentioned clearly. It is not very understandable which one is the Collagen type-I group and which one is the Hyaluronic acid-carboxymethylcellulose group. The used statistic method is not also referred in this table.
Comment 5: In table 3 the complications have to be presented more clearly. It seems to be in total n=4 instead of n=2 complications.
Comment 6: As the authors mention, there is not a direct measurement for the stability of the two compared agents. As a consequence, I believe that there are not enough data to compare the stability of the two agents (as mentioned in the conclusion).
Comment 7: The exact diameters of the used trocars have to be mentioned in the materials and methods section.
Reviewer 2 Report
I would like to command the author team for great effort studying the effectiveness of anti-adhesion agents and I have some questions to address:
1) not clear from the introduction section why did you compare specifically these two agents and it would be nice to have control group.
2) it seems like your study was "non-inferiority type", and if yes it might be better to address it in the materials and methods section
3) why did you use ultrasound method and not MRI, which is according to https://doi.org/10.1016/j.ejrad.2020.108922 is better than US
4) it would be much more clear to present study design according to CONSORT and V points sound a bit confusing throughout the manuscript
5) Did you calculate cost differences as well?
